# Deep-Learning-Based Computer-Aided Systems for Breast Cancer Imaging: A Critical Review

**Yuliana Jiménez-Gaona** [1,2,3,*] **, María José Rodríguez-Álvarez** [2] **and**
**Vasudevan Lakshminarayanan** [3,4,*]

1   Departamento de Química y Ciencias Exactas, Universidad Técnica Particular de Loja,
   San Cayetano Alto s/n CP1101608, Loja, Ecuador
2   Instituto de Instrumentacion para la Imagen Molecular I3M, Universitat Politécnica de Valencia,
   E-46022 Valencia, Spain; mjrodri@i3m.upv.es
3   Theoretical and Experimental Epistemology Lab, School of Optometry and Vision Science,
   University of Waterloo, Waterloo, ON N2L3G1, Canada
4   Department of Systems Design Engineering, Physics, and Electrical and Computer Engineering,
   University of Waterloo, Waterloo, ON N2L3G1, Canada
*   Correspondence: ydjimenez@utpl.edu.ec (Y.J.-G.); vengulak@uwaterloo.ca (V.L.)

**Abstract:** This paper provides a critical review of the literature on deep learning applications in breast tumor diagnosis using ultrasound and mammography images. It also summarizes recent advances in computer-aided diagnosis/detection (CAD) systems, which make use of new deep learning methods to automatically recognize breast images and improve the accuracy of diagnoses made by radiologists. This review is based upon published literature in the past decade (January 2010–January 2020), where we obtained around 250 research articles, and after an eligibility process, 59 articles were presented in more detail. The main findings in the classification process revealed that new DL-CAD methods are useful and effective screening tools for breast cancer, thus reducing the need for manual feature extraction. The breast tumor research community can utilize this survey as a basis for their current and future studies.

**Keywords:** breast cancer; computer-aided diagnosis; convolutional neural networks; deep learning; mammography; ultrasound

## 1. Introduction

Due to the anatomy of the human body, women are more vulnerable to breast cancer than men. Breast cancer is one of the leading causes of death for women globally [1–4] and is a significant public health problem. It occurs due to the uncontrolled growth of breast cells. These cells usually form tumors that can be seen from the breast area via different imaging modalities.

To understand breast cancer, some basic knowledge about the normal structure of the breast is important. Women's breasts are constructed of lobules, ducts, nipples, and fatty tissues (Figure 1) [5]. Normally, epithelial tumors grow inside the lobes, as well as in the ducts, and later form a lump [6], generating breast cancer.

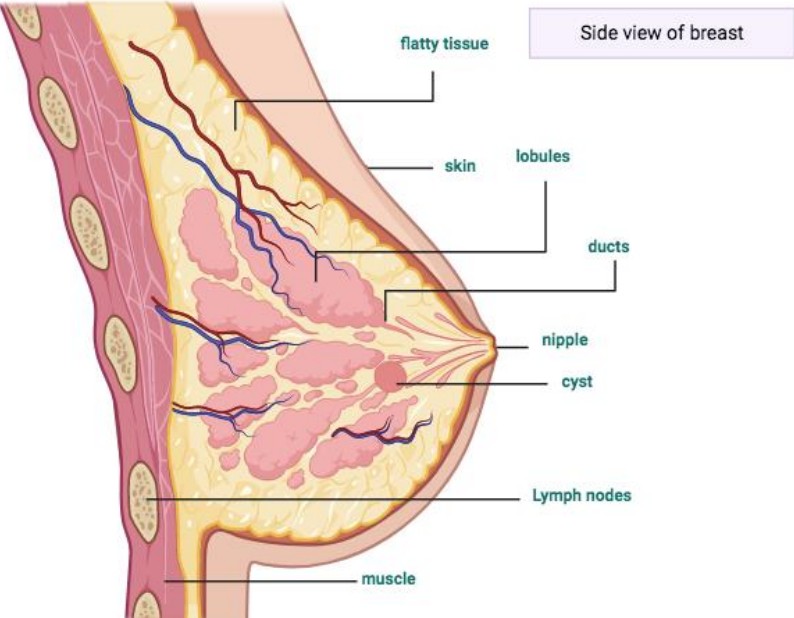

**Figure 1.** This scheme represents the anatomy of a woman's breast. Inside the lobes are the zones where the epithelial tumors or cyst grow. Designed by Biorender (2020). Retrieved from https://app.biorender.com/biorender-templates.

Breast abnormalities that can indicate breast cancer are masses and calcifications [7]. Masses are benign or malignant lumps and can be described in terms of their shape (round, lobular, oval, and irregular) or their margin (obscured, indistinct, and spiculated) characteristics. The spiculated masses are the particular kind of masses that have a high probability of malignancy. A spiculated mass is a lump of tissue with spikes or points on the surface. It is suggestive but not a confirmation of malignancy. It is a common mammography finding in breast carcinoma [8].

On the other hand, microcalcifications are small granular deposits of calcium and may reveal themselves as clusters or patterns (like circles or lines) and appear as bright spots in a mammogram. Benign calcifications are usually larger and coarser with round and smooth contours. Malignant calcifications tend to be numerous, clustered, small, varying in size and shape, angular, and are irregularly shaped [7,9].

Breast cancer screening aims to detect benign or malignant tumors before the symptoms appear, and hence reduce mortality through early intervention [2]. Currently, there are different screening methods, such as mammography [10], magnetic resonance imaging (MRI) [11], ultrasound (US) [12], and computed tomography (CT) [13]. These methods help to visualize hidden diagnostic features. Out of these modalities, ultrasound and mammograms are the most common screening methods for detecting tumors before they become palpable and invasive [2,14–16]. Furthermore, they may be utilized effectively to reduce unnecessary biopsies [17]. These two are the modalities that are reviewed in this article.

A drawback in mammography is that the results depend upon the lesion type, the age of the patient, and the breast density [18–24]. In particular, dense breasts that are "radiographically" hard to see exhibit a low contrast between the cancerous lesions and the background [25,26].

Digital mammography (DM) has some limitations, such as low sensitivity, especially in dense breasts, and therefore other modalities, such as US, are used [12]. US is a non-invasive, non-radioactive, real-time imaging technique that provides high-resolution images [27]. However, all these techniques require manual interpretation by an expert radiologist. Normally, the radiologists try to do a manual interpretation of the medical image via a double mammogram reading to enhance the accuracy of the results [28]. However, this is time-consuming and is highly prone to mistakes [3,29]. Because of

these limitations, different artificial intelligence algorithms are gaining attention due to their excellent performance in image recognition tasks.

Different breast image classification methods have been used to assist doctors in reading and interpreting medical images, such as traditional computer-aided diagnosis/detection (CAD) systems [8,30–32] based on machine learning (ML) [33–35], or based on modern CAD-deep learning (DL) system [36–42].

The goal of CAD is to increase the accuracy and sensitivity rates to support radiologists in their diagnosis decisions [43,44]. Recently, Gao et al. [45] developed a CAD system for screening mammography readings that demonstrated an approximately 92% accuracy in the classification. Likewise, other studies [46,47] used several CNNs for mass detection in mammography's and ultrasounds [48–50].

In general, DL-CAD systems focus on CNNs, which is the most popular model used for intelligent image analysis and for detecting cancer with good performance [51,52]. With CNNs, it is possible to automate the feature extraction process as an internal part of the network, thus minimizing human interference. DL-CAD systems have added broader meaning with this approach, distinguishing it from traditional CAD methods.

The next-generation technologies based on the DL-CAD system solve problems that are hard to solve with traditional CAD [12,33]. These problems include learning from complex data [53,54], image recognition [55], medical diagnosis [56,57], and image enhancement [58]. In using such techniques, the image analysis includes preprocessing, segmentation (selection of a region of interest—ROI), feature extraction/selection, and classification.

In this review, we summarize recent upgrades and improvements in new DL-CAD systems for breast cancer detection/diagnosis using mammograms and ultrasound imaging and then describe the principal findings in the classification process. The following research questions were used as the guidelines for this article:

- How the new DL-CAD systems provide breast imaging classification in comparison with the traditional CAD system?
- Which artificial neural networks implemented in DL-CAD systems give better performance regarding breast tumor classification?
- Which are the main DL-CAD architectures used for breast tumor diagnosis/detection?
- What are the performance metrics used for evaluating DL-CAD systems?

## 2. Materials and Methods

### 2.1. Flowchart of the Review

The research process is shown in Figure 2, which was in accordance with the PRISMA (Preferred reporting items for systematic reviews and meta-analyses) flow diagram and protocol [59].

Furthermore, the systematic review process follows the flow diagram and protocol (Figure 3) given in [60].

We identified appropriate studies in PubMed, Medline, Google Scholar, and Web of Science databases, as well as conference proceedings from IEEE (Institute of Electrical and Electronics Engineers), MICCAI (Medical Image Computing and Computer Assisted Intervention), and SPIE (Society of Photographic Instrumentation Engineers), published between January 2010 and January 2020. The search was designed to identify all studies in which DM and US were evaluated as a primary detection modality for breast cancer, and were both used for screening and diagnosis. A comprehensive search strategy including free text and MeSH terms was utilized, including terms such as: "breast cancer", "breast tumor", "breast ultrasound", "breast diagnostic", "diagnostic imaging", "deep learning", "CAD system", "convolutional neural network", "computer-aided detection", "computer-aided diagnoses", "digital databases", "mammography", "mammary ultrasound", "radiology information", and "screening".

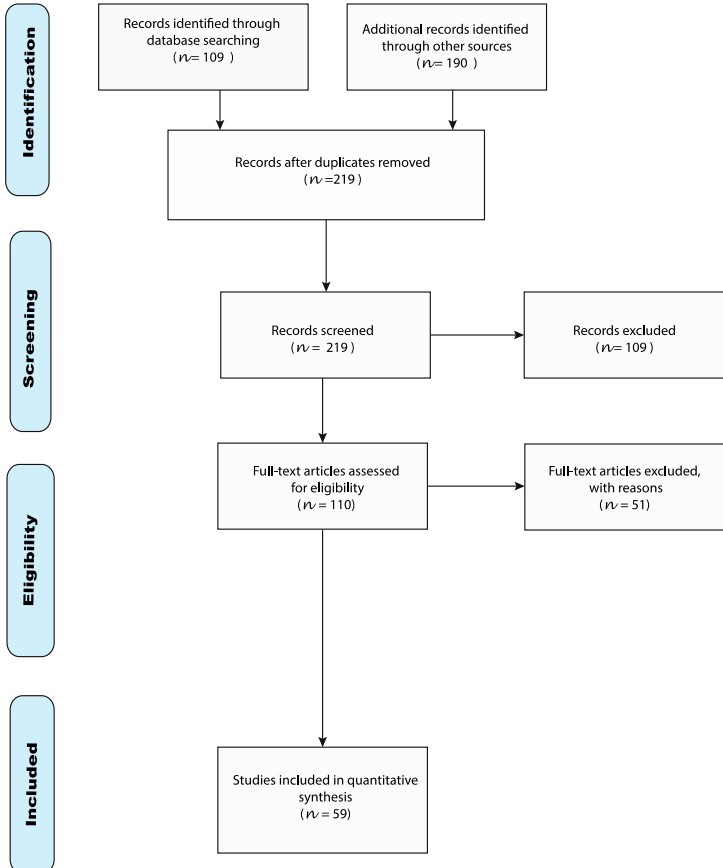

**Figure 2.** PRISMA flow diagram.

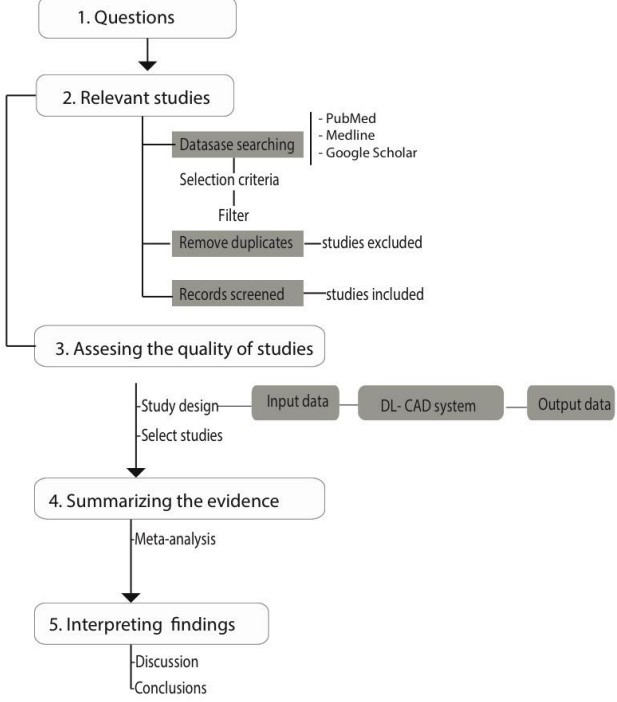

**Figure 3.** This flowchart diagram represents the review process of articles in this paper. DL-CAD: deep learning computer-aided diagnosis/detection.

### 2.1.1. Inclusion Criteria

Articles were included if they assessed computer-aided diagnosis (CADx) and/or computer-aided detection (CADe) for breast cancer, DL in breast imaging, deep CNN, DL in mass segmentation and classification in both DM and US, deep neural network architecture, transfer learning, and feature-based methods regarding automated DM breast density measurements. From a review of the abstracts, we manually selected the relevant papers.

### 2.1.2. Exclusion Criteria

Articles were excluded if the study population included other screening methods, such as MRI, CT, PET (positron emission tomography), or if other machine learning techniques were used.

### 2.2. Study Design

The general modern DL-CAD design was divided into four sections (Figure 4). First, different mammography and ultrasound public digital databases were analyzed as input data for the DL-CAD system. The second section includes the preprocessing and postprocessing in the next-generation DL-CAD.

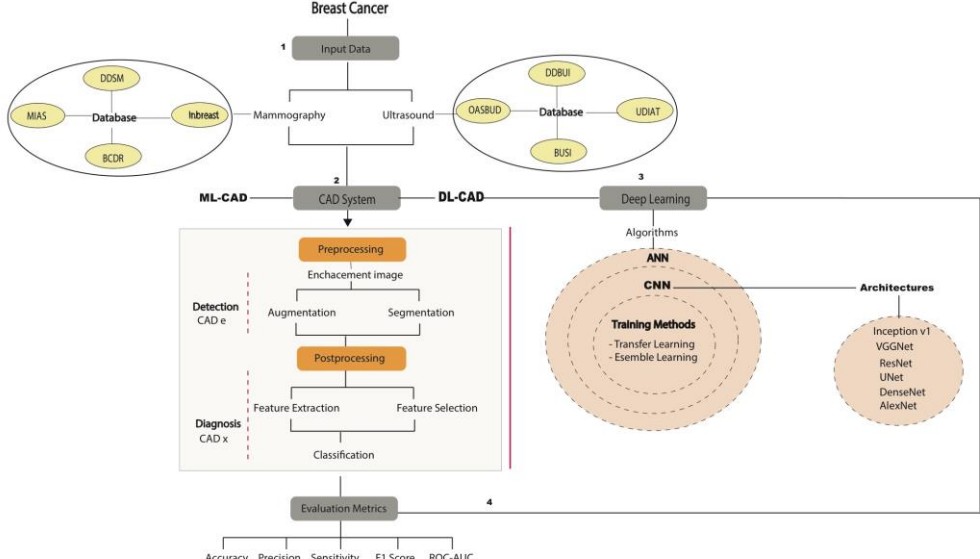

**Figure 4.** The general diagram is a flowchart that describes how a modern CAD system process can be used with DM and US images from public and private databases. Normally, the CAD system consists of several stages, such as segmentation, feature extraction/selection, and classification. However, DL-CAD systems are based on CNN models and architectures for automatic feature extraction/selection and classification with convolutional and fully connected layers through self-learning. Finally, CAD systems are validated by different metrics. ANN: artificial neural network, BCDR: Breast Cancer Digital Repository, BUSI: Breast Ultrasound Image Dataset, CADe: computer-aided detection, CADx: computer-aided diagnosis, DDBUI: Digital Database for Breast Ultrasound Images, DDSM: Digital Database for Screening Mammography, MIAS: Mammographic Image Analysis Society Digital Mammogram Database, OASBUD: Open Access Series of Breast Ultrasonic Data, ROC–AUC: receiver operating characteristic curve–area under the curve, UDIAT: Ultrasound Diagnostic Ultrasound Centre of the Parc Tauli, VGGNet: Visual Geometry Group.

In the third part, full articles were analyzed to compile the successful CNNs used in DL architectures. Furthermore, the best evaluation metrics were analyzed to measure the accuracy of these algorithms. Finally, a discussion and conclusions about these classifiers are presented.

2.2.1. Public Databases

Normally, DL models are tested using private clinical images or publically available digital databases that are used by researchers in the breast cancer area. The amount of public medical images is increasing because most of the DL-CAD systems require a large amount of data. Thus, DL algorithms are applied to available digitized mammograms, such as those from MIAS (Mammographic Image Analysis Society Digital Mammogram Database) [61], DDSM (Digital Database for Screening Mammography), IRMA (Image Retrieval in Medical Application) [62,63], INbreast [64], and BCDR (Breast Cancer Digital Repository) [45,65], as well as public US databases, such as BUSI (Breast Ultrasound Image Dataset), DDBUI (Digital Database for Breast Ultrasound Images), and OASBUD (Open Access Series of Breast Ultrasonic Data) from the Oncology Institute in Warsaw, Poland, and the private US collected datasets, such as SNUH (Seoul National University Hospital, Korea) [48], Dataset A (collected in 2001 from a professional didactic media file for breast imaging specialists) [66], and Dataset B collected from the UDIAT(Ultrasound Diagnostic Ultrasound Centre of the Parc Tauli) Corporation, Sabadell, Spain. These widely used datasets are listed in Table 1.

**Table 1.** Summary of the most commonly used public breast cancer databases in the literature.

| Type | Database | Annotations | Link | Author |
|------|----------|-------------|------|--------|
| Mammograms | DDSM | 2620 patients including mediolateral oblique (MLO) and craniocaudal (CC) images for classification. | http://www.eng.usf.edu/cvprg/Mammography/Database.html | Jiao et al. [67] |
| | BCDR | 736 biopsy prove lesion of 344 patients, including CC and MLO images for classification. | https://bcdr.eu/ | Arevalo et al. [68] |
| | INbreast | 419 cases, including CC and MLO images of 115 patients, for detection and diagnosis. | http://medicalresearch.inescporto.pt/breastresearch/index.php/Get_INbreast_Database | IMoreira et al. [64] |
| | Mini-MIAS | 322 digitized MLO images of 161 patients for segmentation, detection, and classification. | http://peipa.essex.ac.uk/info/mias.html | Peng et al. [69] |
| Ultrasound | BUSI | The dataset consists of 600 female patients. The 780 images include 133 normal images without masses, 437 images with cancer masses, and 210 images with benign masses. This set is utilized for classification, detection, and segmentation. | https://scholar.cu.edu.eg/?q=afahmy/pages/dataset | Dhabyani et al. [70] |
| | DDBUI | 285 cases and 1132 images in total for classification. | https://www.atlantis-press.com/proceedings/jcis2008/1735 | Tian et al. [71] |
| | Dataset A | Private dataset with 306 (60 malignant and 246 benign) images, which are utilized for detection. | goo.gl/SJmoti | Yap et al. [48] |
| | Dataset B | Private dataset with 163 (53 malignant and 110 benign) images. | | Byra et al. [66] |
| | SNUH | Private dataset with a total of 1225 patients with 1687 tumors. This study includes biopsy diagnosis. | | Moon et al. [49] |
| | OASBUD | 52 malignant and 48 benign masses, which are utilized in image processing algorithms. | http://bluebox.ippt.gov.pl/~hpiotrzk | Piotrzkowska et al. [72] |
| | ImageNet | 882 US images (678 benign and 204 malignant lesions), which are utilized in object recognition, image classification, and automatic object clustering. | http://www.image-net.org/ | Deng et al. [73] |

SNUH: Seoul National University Hospital.

### 2.2.2. CAD Focused on DM and US

The CAD systems are divided into two categories. One is the traditional CAD system and the other is the DL-CAD system (Figure 5). In the traditional CAD system, the radiologist or clinician defines features in the image, where there can be problems regarding recognizing the shape and density information of the cancerous area. A DL-CAD system, on the other hand, creates such features by itself through the learning process [74].

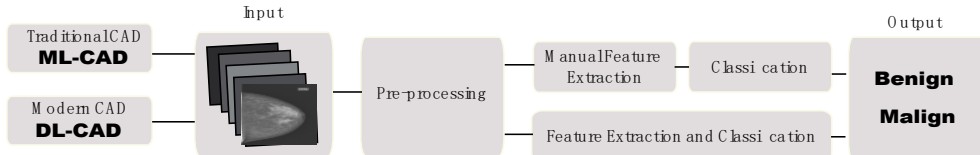

**Figure 5.** The scheme describes the main difference between the traditional machine learning (ML)-CAD system and the DL-CAD system.

Furthermore, CAD systems can be broken down into two main groups: CADe and CADx. The main difference between CADe and CADx is that the first refers to a software tool that assists in ROI segmentation within an image [75], identifying possible abnormalities and leaving the interpretation to the radiologist [8]. On the other hand, CADx serves as a decision aid for radiologists to characterize findings from a CADe system. Several significant CAD works are described in Table 2.

**Table 2.** The traditional CAD system summary with DM and US breast cancer images. It covers four stages: (i) image processing, (ii) segmentation, (iii) feature extraction and selection, and (iv) classification.

| Reference | Models | Description | Application |
|---|---|---|---|
| [76,77] | Pixel-based, which is based on the curvature of the edge and clustering [3,78]: conventional (CLAHE), region-based, feature-based (wavelet), and fuzzy. | Pectoral removal techniques are not sufficient to provide accurate results. Thereby, intensity-based methods, line detection, statistical techniques, wavelet methods, and the active contour technique have also been tried for segmenting this area. Its accuracy varies from 84% to 99%, where the active contour technique reached the highest value of 99%, followed by the wavelet method with 93% [79]. Enhancement techniques are divided into three categories: spatial, frequency domain, and a mixture of these two. These categories can be classified into four models. The region-based method requires a seed point and it is time-consuming. | Preprocessing |
| [80,81] | Local thresholding and region-growing [82]; edge detection, template matching [12,83], and a multiscale technique [84]; NN [85]. | The thresholding method shows greater stability but is dependent on the parameter selection. Furthermore, is not sufficient for segmenting fatty tissue in a DM because its images contain noise and have low contrast and intensity. The region-growing method is well-known in micro-calcification detection and uses pixel properties for segmenting fatty tissue. Edge detection utilizes the wavelet transform in a multiscale structure to represent signals and variations in US images. Template matching requires a comparison with a given image (ROI) with a template image to measure the similarity between both. Finally, an NN utilizes a multi-layered perceptron with a hidden layer for extracting the contours of tumors automatically; nevertheless, training an NN is time-consuming. | Segmentation |

**Table 2.** *Cont.*

| Reference | Models | Description | Application |
|---|---|---|---|
| [86] | PCA [87], LDA [88], and GLCM [89]. | Feature selection methods: wrapper and filter (chi-square [90]). The most well-known feature extraction techniques are PCA, LDA, GLCM, gain ratio, recursive feature [91], RF, WPT [92,93], Fourier power spectrum [94], Gaussian [95] and DBT [29]. PCA feature extraction techniques are better at reducing the high-dimensional correlated features into low dimensional features [87]. | Feature Selection and extraction |
| [12,33] | SVM [96,97] and ANN [98,99]. | SVM is useful in DM classification because these are highly overlapping and nonlinear in their feature space. It minimizes the generalization error during the process of testing data and is much more accurate and computational efficient because of the reduced number of parameters. ANN: Backpropagation, SOM, and hierarchical ANN. The performance of back-propagation is better than that of linear classifiers. However, the training process is stochastic and unrepeatable, even with the same data and initial conditions. Prone to overfitting due to the complexity of the model structure. Finally, advantages and disadvantages from other classifiers have been previously discussed in several studies: KNN [100], BDT [101], simple logistic classifier [102], and DBN [103] | Classification |

### 2.2.3. Preprocessing

It is known that the database characteristics can significantly affect the performance of a CAD scheme, or even a particular processing technique. Furthermore, it can develop a scheme that yields erroneous or confusing results [104] since radiological images contain noise, artefacts, and other factors that can affect medical and computer interpretations. Thus, the first step in preprocessing is to improve the image quality, contrast, removal noise, and pectoral muscle [105].

Image Enhancement

The main purpose of image preprocessing is to enhance the image and suppress noise while preserving important diagnostic features [106,107]. Preprocessing for breast cancer diagnosis also consists of delineation of the tumors from the background, breast border extraction, and pectoral muscle removal. The pectoral muscle segmentation is a challenge in mammogram image analysis because the density and texture information is similar to that of the breast tissues. Furthermore, it depends on the standard view used during mammography. Generally, mediolateral oblique (MLO) and craniocaudal (CC) views are used [78].

As noted, DM includes many sources of noise, which are classified as a high-intensity, low-intensity, or tape artefacts. The principal noise models observed in mammography are salt and pepper, Gaussian, speckle, and Poisson noise.

In the same way, US images suffer from noise, such as intensity inhomogeneity, a low signal-to-noise ratio, high speckle noise [108,109], blurry boundaries, shadow, attenuation, speckle interference, and low contrast. Speckle noise reduction techniques are categorized in filtering, wavelet, and compound methods [12].

Thus, many traditional filters can be applied for noise removal, including a wavelet transform, median filter, mean filter, adaptive median filter, Gaussian filter, and adaptive Wiener filter [3,110–113]. Furthermore, different traditional methods, such as histogram equalization (HE) [114,115],

adaptive histogram equalization (AHE) [116], and contrast-limited adaptive histogram equalization (CLAHE) [117], can be used to enhance the image.

On the other hand, deep CNNs [118] are gaining attention for improving super-resolution [119] images (SR) based on a CNN, namely, (i) multi-image super-resolution and (ii) single-image super-resolution [120,121]. Among the most used algorithms for generating high-resolution (HR) imaging [122,123] are nearest-neighbor interpolation [124], bilinear interpolation [125], and bicubic interpolation [126].

### Image Augmentation

Deep CNN depends on large datasets to avoid overfitting and is necessary for good DL model performance [127]. Thus, limited datasets are a major challenge in medical image processing [128] and it is necessary to implement data augmentation techniques. There are two common techniques for increasing the data in DL, namely, data augmentation and transfer learning/fine-tuning [129,130]. Examples of DL models that have been trained with data augmentation are Imagenet [74] and transfer learning [47].

The image augmentation algorithms include basic image manipulations (flipping, rotation, transformation, feature space augmentation, kernel, mixing images, and random erasing [131]) and DL manipulations (generative adversarial networks (GANs)) [132], along with a neural style transfer [133] and meta-learning [128]). These techniques increase the amount of data by preprocessing input image data via operations such as contrast enhancement and noise addition, which have been implemented in many studies [134–140].

### Image Segmentation

This processing step plays an important role in image classification. Segmentation is the separation of ROIs (lesions, masses, and microcalcifications) from the background of the image.

In traditional CAD systems, the tasks of specifying the ROI, such as an initial boundary or lesions, are accomplished with the expertise of radiologists. The traditional segmentation task in DM can be divided into four main classes: (i) threshold-based segmentation, (ii) region-based segmentation, (iii) pixel-based segmentation, and (iv) model-based segmentation [3,78]. Furthermore, US image segmentation includes several techniques: threshold-based, region-based, edge-based, water-based, active-contour-based, and neural-network-learning-based techniques [141,142].

The accuracy of the segmentation affects the results of CAD systems because numerous features are used for distinguishing malignant and benign tumors (texture, contour, and shape of lesions). Thus, the features may only be effectively extracted if the segmentation of tumors is performed with great accuracy [106,142]. This is why researchers are using DL methods, especially CNNs, because these methods have shown excellent results on segmentation tasks. Furthermore, DL-CAD systems are independent of human involvement and are capable of autonomously modeling breast US and DM knowledge using constraints. Two strategies have been utilized for full image sizes for training CNNs for DM and US instead of ROIs: (1) high-resolution [143] and (2) patch-level [144] images. For example, recent network architectures that have been used to produce segmented regions are YOLO [145], SegNet [146,147], UNet [148], GAN [149], and ERFNet [150].

### 2.2.4. Postprocessing

### Image Feature Extraction and Selection

After the segmentation, feature extraction and selection are the next steps to remove the irrelevant and redundant information of the data being processed. Features are characteristics of the ROI taken from the shape and margin of lesions, masses, and calcifications. These features can be categorized into texture and morphologic features [12,86], descriptors, and model-based features [52], which help

to discriminate between benign and malignant lesions. Most of the texture features are calculated from the entire image or ROIs using the gray-level value and the morphological features.

There are some traditional techniques used for feature selection, such as searching algorithms, the chi-square test, random forest, gain ratio, and recursive feature elimination [91]. In addition, other traditional techniques used for the feature extraction include principal component analysis (PCA), wavelet packet transform (WPT) [92,93], grey-level co-occurrence matrix (GLCM) [91], Fourier power spectrum (FPS) [94], Gaussian derivative kernels [95], and decision boundary features (DBT) [151].

However, in some classification processes, such as an ANN or support vector machine (SVM), the dimension of the vectors affects both the computational time and the performance [152] because this depends on the number of features extracted. Thus, feature selection techniques reduce the size of the feature space, improving the accuracy and computation time by eliminating redundant features [153]. In particular, DL models produce a set of image features from the data [154], whose main advantage is that they extract features and perform classifications directly. Providing good extraction and selection of the features is a crucial task for DL-CAD systems; for example, some CNNs that are capable of extracting features have been presented by different authors [155,156].

### 2.2.5. Classification

During the classification process, the dimension of feature vectors is important because these affect the performance of the classifier. The features of breast US images can be divided into four types: texture, morphological, model-based, and descriptor features [86]. After the features have been extracted and selected, they are input into a classifier to categorize the ROI into malignant and benign classes. The commonly used classifiers include linear, ANN, Bayesian neural networks, decision tree, SVM, template matching [106], and CNNs.

Recently, deep CNNs, which are hierarchical architectures trained on large-scale datasets, have shown stunning performances regarding object recognition and detection [157], which suggests that these could also improve breast lesion detection in both US and DM methods. Some researchers are interested in lesion [158], microcalcification [159,160], and mass [161,162] classification in DM and US [15–154,163–165] images based on CNN models.

### Deep Learning Models

DL in medical imaging is mostly represented by a basic structure called a CNN [57,75]. There are different DL techniques, such as GANs, deep autoencoders (DANs), restricted Boltzmann machine (RBM), stacked autoencoders (SAEs), convolutional autoencoders (CAEs), recurrent neural networks (RNNs), long short-term memory (LSTM), multiscale convolutional neural network (M-CNN), and multi-instance learning convolutional neural network (MIL-CNN) [3]. DL techniques have been implemented to train neural networks for breast lesion detection, including ensemble [75] and transfer learning [129,157,166] methods. The ensemble method combines several basic models in order to get an optimal model [167], and transfer learning is an effective DL method to pre-train models to deal with small datasets, as in the case of medical images.

ANNs are composed of an input and output layer, plus one or more hidden layers, as shown in Figure 6. In the field of breast cancer, three types of ANN are frequently used: backpropagation, SOM, and hierarchical ANNs. To train an ANN with a backpropagation algorithm, the error function is given to calculate the gradient descent. This error propagates in the backward direction and the weights are adjusted for error reduction. This processing is repeated until the error becomes zero or is a minimum [3].

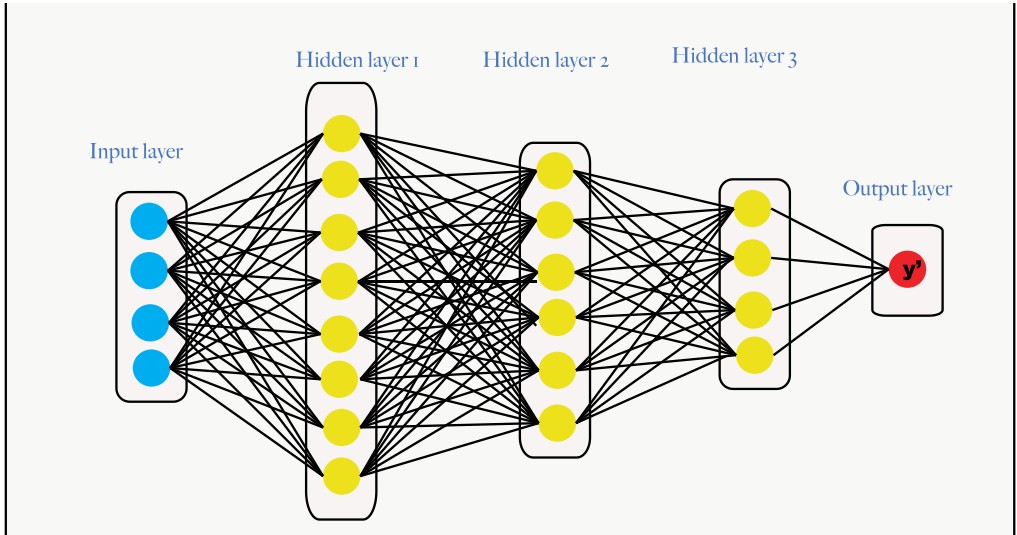

**Figure 6.** An ANN learns by processing images, where each of which contains an input, hidden, and result layer.

Convolutional Neural Networks

CNNs are the most widely used Neural Networks when it comes to DL and medical image analysis. The CNN structure has three types of layers: (i) convolution, (ii) pooling, and (iii) full-connection layers, which are stacked in multiple layers [74]. Thus, a CNN's structure is determined by different parameters, such as the number of hidden layers, the learning rate, the activation function (rectified linear unit (ReLU)), pooling layer for feature map extraction, loss function (softmax), and the fully connected layers for classification, as shown in Figure 7.

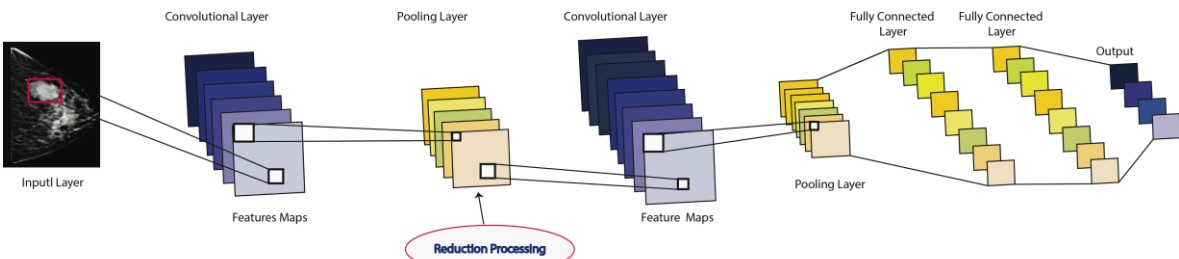

**Figure 7.** A feed-forward CNN network, where the convolutional layers are the main components, followed by a nonlinear layer (rectified linear unit (ReLU)), pooling layer for feature map extraction, loss function (softmax), and the fully connected layers for classification. The output can be either benign or malignant classes.

Furthermore, there are several methods for improving a CNN's performance, such as dropout and batch normalization. Dropout is a regularization method that is used to prevent a CNN model from overfitting. A batch normalization layer speeds up the training of CNNs and reduces the sensitivity to network initialization.

2.2.6. Evaluation Metrics

Different quantitative metrics are used to evaluate the classifier performance of a DL-CAD system. These include accuracy (Acc), Sensitivity (Sen), Specificity (Spe), area under the curve (AUC), F1 score, and a confusion matrix. The statistical equations are shown in Tables 3 and 4.

**Table 3.** Confusion matrix for a binary classifier that is used to distinguish between two classes, namely, benign and malignant. TP: true positive; FN: false negative, FP: false positive, TN: true negative, TPR: true positive rate, FPR: false positive rate.

| Classes | Predicted Classes | | Equation |
|---|---|---|---|
| | $C_1$ | $C_2$ | |
| $C_1$ (Benign) | TP | FN | $\text{TPR} = \left(\dfrac{\text{TP}}{\text{TP} + \text{FN}}\right)$ |
| $C_2$ (Malignant) | FP | TN | $\text{FPR} = \left(\dfrac{\text{FP}}{\text{FP} + \text{TN}}\right)$ |

**Table 4.** Validation assessment measures.

| Model | Equation |
|---|---|
| Accuracy | $\text{Acc} = \left(\dfrac{\text{TP} + \text{TN}}{\text{TP} + \text{TN} + \text{FP} + \text{FN}}\right)$ |
| Sensitivity | TPR |
| Specificity | $\text{TNR} = \left(\dfrac{\text{TN}}{\text{TN} + \text{FN}}\right)$ |
| Precision | $\text{Precision} = \left(\dfrac{\text{TP}}{\text{TP} + \text{FP}}\right)$ |
| F1 Score | $\text{F1 Score} = 2 \times \left(\dfrac{\text{precision} \times \text{recall}}{\text{precision} + \text{recall}}\right)$ |
| MCC | $\text{MCC} = \dfrac{\text{TP} \times \text{TN} - \text{FP} \times \text{FN}}{\sqrt{(\text{TP} + \text{FP})\,(\text{TP} + \text{FN})\,(\text{TN} + \text{FP})\,(\text{TN} + \text{FN})}}$ |

The receiver operating characteristic curve (ROC) is a graph for plotting the true positive rate (TPR) versus a false positive rate (FPR) and is derived from the AUC. The TPR and the FPR are also called sensitivity (recall) and specificity, respectively, as shown in Figure 8.

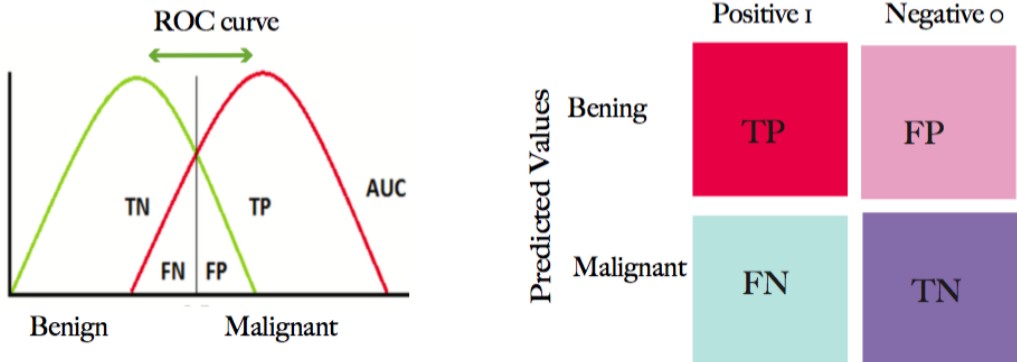

**Figure 8.** The confusion matrix for the ROC. The number of images correctly predicted by the classifier is located on the diagonal. The ROC curve utilizes the TPR on the *y*-axis and the FPR on the *x*-axis.

The AUC provides the area under the ROC curve and a perfect score has a range from 0.5 to 1. A 100% correct classified version will have an AUC value of 1 and it will be 0 if there is a 100% wrong classification [168].

Cross-validation is a statistical technique that is used to evaluate predictive models by partitioning the original samples into training, validation, and testing sets. There are three types of validation: (1) hold-out splits (training 80% and testing 20%), (2) three-way data split (training 60%, validation 20%, and testing 20%), and (3) K-fold cross-validation (from 3 to 5 k-fold for a large data set, 10 k-fold for a small dataset), where the data are split into k different subsets depending on their size [65].

## 3. Results

### 3.1. CNN Architectures

A model's performance depends on the architecture and the size of the data. In this sense, there are different CNN architectures that have been proposed: AlexNet [169], VGG-16 [170], ResNet [171], Inception (GoogleNet) [172], and DenseNet [173]. These networks have shown promising performance in recent works for image detection and classification. Table 5 shows a brief description of these networks.

**Table 5.** Summary of CNN architecture information for breast imaging processing.

| Reference | Model | Description | Training Method | Application |
|---|---|---|---|---|
| Krizhevsky et al. [169] | AlexNet | A deep CNN evaluated using the Imagenet [65] LSVRC-2010 dataset [173], with top-1 and top-5 error rates of 37.5% and 17.0%, respectively. This achieved a top-5 test error rate of 15.3% compared to 26.2% (ImageNet Large-Scale Visual Recognition Challenge (ILSVRC) 2012). | Dropout model | Classification |
| Samala et al. [174] | DL-CNN | CAD system for masses in DBT volume, which is trained using transfer learning. The best AUC obtained was 0.933 and the improvement was statistically significant ($p < 0.05$). | CNN architecture | Detection tomosynthesis from DM |
| Simoyan et al. [170] | VGG-VD | The very deep (VD)-CNN models (VGG-VD16 and VGG-VD19 [158]) were evaluated in ILSVRC 2014 (ImageNet). | Deep ConvNet architecture | Classification |
| He et al. [171] | ResNet | An ensemble of these residual nets achieved a 3.57% error on the ImageNet (ILSVRC 2015) test set. | ResNet with a depth of up to 152 layers 8× deeper | Classification |
| Huang et al. [172] | DenseNet | DenseNet was proposed to reduce the vanishing gradient problem, to reduce the number of parameters, and to strengthen the feature propagation. | ImageNet with a CNN | Object recognition |
| Szegedy et al. [27] | Inception v5 | A deep CNN was evaluated in ILSVRC 2014. | Deep-CNN | Classification and detection |
| Das et al. [175] | VGGNet | BreakHist dataset with 58 malignant and 24 benign cases was evaluated with a deep CNN. The best accuracy percentage was reached with 100× (89.06%). | MIL architecture | Histopathology |
| Cao et al. [152] | Deep CNN | Private dataset that contains 577 benign and 464 malignant cases. | Detection: Fast R-CNN, Faster R-CNN, YOLOv3, and SSD; Classification: AlexNet, VGG, ResNet, GoogleNet, ZFNet, and Densenet | US lesion detection and classification |
| Chiao et al. [153] | Deep CNN | Private US imaging dataset that contains 307 images with 107 benign and 129 malignant cases. | Mask R-CNN with ROI alignment; based on a faster R-CNN using an RPN to extract features | Sonogram lesion detection and classification |
| Yap et al. [48] | LeNet, UNet, deep CNN | This work studies the performance of CNNs in breast US detection using two private datasets A and B. | LeNet [163], U-Net [148], and transfer learning [176] | US breast lesion detection |
| Geras, K. et al. [176] | Multi-view DL-CNN | INbreast [77] and DDSM [58] databases were used; the model achieved an AUC of 0.68%. | The CNN is jointly trained using stochastic gradient descent with backpropagation [175,176] and data augmentation via random cropping [168,177] | High-resolution, augmentation, and DM classification |

**Table 5.** *Cont.*

| Reference | Model | Description | Training Method | Application |
|---|---|---|---|---|
| Han et al. [62] | GoogleNet with ensemble learning | Dataset contains a total of 7408 US breast images, with 657 used as the training set and 829 as the test set. The accuracy reached was 90.21%. | The CNN was trained with 10-fold cross-validation. Data augmentation was carrying out with the Caffe method | Data augmentation, detection, and classification of breast lesions in US |
| Dhungel et al. [178] | LeNet for CNN models in cascade R-CNN | INbreast dataset was used, with 115 cases and 410 images from MLO and CC views. The results showed that the DL-CAD system is able to detect 90% of masses, with a segmentation accuracy of 85% and the classification reached a sensitivity of 0.98 and a specificity of 0.7. | DL detection: Fast R-CNN, multiscale-DBN, and random forest; DL segmentation: CRF; DL classification: regression method. | Detection, segmentation, and classification of masses in DM |
| Singh et al. [165] | GAN | The Mendeley database [179] was used, which contains 150 malignant and 100 benign tumors. The performance metrics achieved scores of dice = 93.76% and IoU = 88.82%. | Segmentation with GAN learning. | Segmentation and classification of US images |
| Cheng, J. Z. [37] | SDAE based CADx | The method was carried out on a private database, with 520 breast sonograms (275 benign and 245 malignant lesions). The AUC performance reached 0.80%. | An SDAE (OverFeat) model was used to classify with the ensemble method. | Breast lesion/nodules diagnosis and classification of US images |

## 3.2. Performance Metrics

Furthermore, brief reviews of the DL architectures based on DM and US breast images, along with their evaluation metrics, are presented in Tables 6 and 7 [50,180].

**Table 6.** The quantitative indicators that were used to evaluate the performance between different CNN architectures in DM datasets.

| Reference | Database | Deep CNN Model | Acc (%) | Sen (%) | Spec (%) | Precision (%) | F1 Score (%) | AUC (%) |
|---|---|---|---|---|---|---|---|---|
| Al-Masni et al. [145] | DDSM with 600 DM. | CNN YOLO5: Fold cross-validation in both datasets; mass classification | 99 | 93.20 | 78 | - | - | 87.74 |
| | DDSM augmentation with 2.400 | Mass detection | 97 | 100 | 94 | - | - | 96.45 |
| Ragab et al. [168] | DDSM with 2620 cases | Deep-CNN-based linear SVM using ROI manually | 79 | 76.3 | 82.2 | 85 | 80 | 88 |
| | CBIS- DDSM with 1644 cases | ROI threshold | 80.5 | 77.4 | 84.2 | 86 | 81.5 | 88 |
| | | SVM-based medium Gaussian | 87.2 | 86.2 | 87.7 | 88 | 87.1 | 94 |
| Duggento et al. [180] | CBIS-DDSM | Deep CNN | 71 | 84.4 | 62.4 | - | - | 77 |
| Chougrad et al. [181] | BCDR | Inceptionv3 | 96.67 | - | - | - | - | 96 |
| | DDSM | | 97.35 | - | - | - | - | 98 |
| | INbreast | | 95.50 | - | - | - | - | 97 |
| | MIAS | | 98.23 | - | - | – | - | 99 |

**Table 7.** The quantitative indicators that were used to evaluate different CNN architectures' performances on US datasets.

| Reference | Database | Deep CNN Model | Acc (%) | Sen (%) | Spec (%) | Precision (%) | F1 Score (%) | AUC (%) |
|---|---|---|---|---|---|---|---|---|
| Moon et al. [49] | BUSI SNUH | VGGNet-like | 84.57 | 73.65 | 93.12 | 89.34 | 80.74 | 91.98 |
| | | VGGNet 16 | 84.57 | 73.64 | 93.12 | 89.34 | 80.74 | 93.22 |
| | | ResNet 18 | 81.60 | 86.49 | 77.77 | 75.29 | 80.50 | 91.85 |
| | | ResNet 50 | 81.60 | 75.68 | 86.24 | 81.16 | 78.32 | 88.83 |
| | | ResNet 101 | 84.57 | 75,00 | 92.06 | 88.10 | 81.02 | 91.04 |
| | | DenseNet 40 | 85.46 | 79.05 | 90.48 | 86.67 | 82.69 | 93.52 |
| | | DenseNet 12 | 86.35 | 77.70 | 93.12 | 89.84 | 83.33 | 92.48 |
| | | DenseNet 161 | 83.09 | 69.59 | 93.65 | 89.57 | 78.33 | 89.18 |
| Byra et al. [66,182] | ImageNet | VGG19 | 88.7 | 0.848 | 0.897 | - | - | 93.6 |
| | UDIAT | | 84 | 0.851 | 0.834 | - | - | 89.3 |
| | OASBUD [150] | | 83 | 0.807 | 0.854 | - | - | 88.1 |
| Cao et al. [152] | Private dataset consisting of 579 benign and 464 malignant cases | Single Shot Detector (SSD)300 + ZFNet | 96.89 | 67.23 | - | - | 79.38 | - |
| | | YOLO | 96.81 | 65.83 | - | - | 78.37 | - |
| | | SSD300 + VGG16 | 96.42 | 66.70 | - | - | 78.85 | - |
| Han et al. [62] | Private database with a total of 7408 US images with 4254 benign and 3154 malignant lesions | CNN-based GoogleNet | 91.23 | 84.29 | 96.07 | | - | 91 |
| Shan et al. [35] | Private database containing 283 breast US images (133 cases are benign and 150 cases are malignant) | ANN | 78.1 | 78 | 78.2 | - | - | 82.3 |

Furthermore, Table 8 gives a brief overview of the new DL-CAD systems' approaches and the traditional ML-CAD systems.

**Table 8.** DL-CAD systems vs. traditional ML-CAD systems.

| Reference | Application | Method | Dataset | Acc (%) | Sen (%) | Spec (%) | AUC (%) | Error (%) |
|---|---|---|---|---|---|---|---|---|
| Dheeba [183] | DM classification | ML wavelet neural network | Private database consisting of 216 multiview CC and MLO images. | 93.67 | 94.16 | 92.10 | 96.85 | 96.85 |
| Trivizakis et al. [184] | DM classification | ML with transfer learning and features based on ImageNet and CNN architecture | Mini MIAS and DDSM | 79.3 | - | - | 84.2 | - |
| | | | | 74.8 | - | - | 78.00 | - |
| Samala et al. [185] | DM classification | Multitask transfer learning by a Deep CNN | ImageNet | 90 | - | - | 82 | - |
| Jadoon et al. [186] | DM extraction and classification | CNN + wavelet | IRMA, DDSM, and MIAS | 81.83 | - | - | 83.1 | 15.43 |
| | | CNN + SVM | | 83.74 | - | - | 83.9 | 17.46 |
| Debelee et al. [42] | DM extraction | CNN + SVM | MIAS | 97.46 | 96.26 | 100 | - | - |
| | | | DDSM | 99 | 99.48 | 98.16 | - | - |
| | | | MIAS | 87.64 | 96.65 | 75.73 | - | - |
| | | MLP | DDSM | 97 | 97.40 | 96.26 | - | - |
| | | | MIAS | 91.11 | 86.66 | 100 | - | - |
| | | KNN + SVM | DDSM | 97.18 | 100 | 95.65 | - | - |
| Ahmed et al. [187] | DM detection | Deep CNN with five–fold cross-validation | INbreast | 80.10 | 80 | - | 78 | - |
| Xu et al. [51] | US image segmentation | Deep CNN | Private 3D breast US | 90.13 | 88.88 | - | - | - |

**Table 8.** *Cont.*

| Reference | Application | Method | Dataset | Acc (%) | Sen (%) | Spec (%) | AUC (%) | Error (%) |
|---|---|---|---|---|---|---|---|---|
| Shan et al. [35] | US image segmentation | ML decision tree | Private breast US consisting of 283 images, where 133 cases are benign and 150 cases are malignant | 77.7 | 74.0 | 82.0 | 80 | - |
| | | ANN | | 78.1 | 78.0 | 78.2 | 82 | - |
| | | Random forest | | 78.5 | 75.3 | 82 | 82 | - |
| | | SVM | | 77.7 | 77.3 | 78.2 | 84 | - |
| Gu et al. [188] | 3D US image segmentation | Preprocessing: morphological reconstruction; segmentation: region-based approach | Private database with 21 cases, with masses prior to biopsy | 85.7 | - | - | - | - |
| Zhang et al. [36] | US image feature extraction and classification | DL architecture | The private dataset consisting of 227 elastography images, with 135 benign tumors and 92 malignant tumors | 93.4 | 88.6 | 97.1 | 94.7 | - |
| Almajalid et al. [147] | US image segmentation | DL-CNN architecture U-net | The private dataset containing 221 BUS images | 82.52 | 78.66 | 18.59 | - | - |
| Singh et al. [189] | US image classification | ML fuzzy c-means and backpropagation ANN | 178 breast US containing 88 benign and 90 malignant cases | 95.86 | 95.14 | 96.58 | 95.85 | - |
| Cheng et al. [37] | US (sonogram) classification | DL-SDAE | 520 breast US (275 benign and 245 malignant lesions) | 82.4 | 78.7 | 85.7 | 89.6 | _ |
| Shi, et al. [190] | US image classification | Deep polynomial network | A total of 200 pathology-proven breast US images | 92.40 | 92.67 | 91.36 | - | - |

## 4. Discussion and Conclusions

Considering that breast tumor screening using DM has some consequences and limitations because a higher number of unnecessary biopsies and ionizing radiation exposure endangers the patient's health [12], along with low specificity and high FP results, which imply higher, recall rates and higher FN results [191]. This is why US is used as the second choice for DM. Thus, US imaging is one of the most effective tools in breast cancer detection because it has been shown to achieve high accuracy in mass detection, classification [38], and diagnosis of abnormalities in dense breasts [192].

For the abovementioned reasons, we have addressed using both kinds (DM and US) of images in this review, focusing on different ML and DL architectures applied in breast tumor processing, and offering a general overview of databases and CNNs, including their relation and efficacy in performing segmentation, feature extraction, selection, and classification tasks [192].

Thus, according to the research shown in Table 1, the most utilized databases for DM images are MIAS and DDSM, and for US image classification, the public databases BUSI, DDBUI, and OASBUD are most used. The DM images contributed to 110 and 168 published conference papers for the DDSM and MIAS databases, respectively [5]. However, the databases report some limitations and advantages; for example, the MIAS database contains a limited number of images, strong noise, and low-resolution images. In contrast, the DDSM contains a big dataset. Likewise, INbreast contains high-resolution images but has a small data size. BCDR, in comparison with DDSM, has been used in a few studies. Some details about the others strengths and limitations of these databases are discussed in Abdelhafiz [65].

Thereby, Table 2 shows a summary of traditional ML-CAD systems that use public and private databases of DM and US breast images. It covers (i) image preprocessing and (ii) postprocessing steps. This is in contrast with Table 5, which shows a brief summary of DL-CAD systems based on CNN architectures in both types of digital breast images. Thus, in Table 5, various DL architectures and their training strategies for detection and classification tasks are discussed. Based on the most popular datasets, CNN seems to perform rather well, as demonstrated by Chiao et al., Yap et al., and Samala et al. [48,153,174]. Furthermore, [169,173] used several preprocessing and postprocessing

techniques for high-resolution [58] data augmentation, segmentation, and classification. The most commonly CNNs used are AlexNet, VGG, ResNet, DenseNet, Inception (GoogleNet), LeNet, and UNet, which employ recent Python libraries for implementing CNNs, such as Tensorflow, Caffe, and Keras, with different hyper-parameters to training the network [55].

Most of these DL architectures use a large data set; thus, it is required to apply an augmentation technique to avoid overfitting and to have better performance during classification. In this sense, the researchers mentioned in Table 6 [145,168,180,181] and Table 7 [35,49,62,66,152,182] the authors used transfer learning and ensemble methods, such as data augmentation, to improve the performance of the CNN network, reaching an 89.86% accuracy and 0.9578% AUC in DM, and an AUC of 0.68% on US images. Furthermore, Singh et al. [165] showed that the results obtained with a GAN for breast tumor segmentation outperformed the UNet model, and the SegNet and ERFNet models yielded the worst segmentation results on US images.

In addition, according to Cheng et al. [37], DL techniques could potentially change the design paradigm of CADx systems due to their several advantages over the traditional CAD systems. These are as follows: First, DL can directly extract features from the training data. Second, the feature selection process will be significantly simplified. Third, the three steps of feature extraction, selection, and classification can be realized within the same deep architecture. Thus, SDAE architecture can potentially address the issues of high variation in either the shape or appearance of lesions/tumors. Furthermore, various studies [39–41,55] prove that those CNN methods that compare images from CC and MLO views improve the accuracy of detection and reduce the FPR.

Furthermore, different evaluation metrics are described in Tables 3 and 4 as corroboration of the performance of these techniques. The results in Tables 6 and 7 describe different research where their authors have used a variety of datasets (Table 1), approaches, and performance metrics to evaluate CNN techniques in DM and US imaging. For example, better results were achieved in DM analysis by Al-Masni [145] with YOLO5 using DDSM data augmentation, while Chougrad et al. [181] used a deep CNN (Inception V3) with DDSM and MIAS datasets. On the other hand, Moon et al. [49] introduced a DenseNet model to analyze private (BUSI and SNUH) US datasets. Byra et al. [66] achieved high accuracy with the VGG19 deep CNN model using the ImageNet database. Similarly, Cao et al. [152] attained an accuracy of 96.89% with SSD + ZFNet and Han et al. [62] reached 91.23% using a private dataset with GoogleNet.

Likewise, Table 8 contains a literature review for the comparison of the evaluation metrics between DL-CAD systems and traditional ML-CAD systems. Even though Table 8 shows that Deheeba et al. [183] presented a good traditional wavelet neural network CAD system with high accuracy (93.67%) and AUC of 96.85%, Debelee et al. [42] exceeded this percentage using a CNN + SVM DL-CAD system with DDSM (99%) and MIAS (97.18%) DM datasets. In US images Zhang et al. [36] and Shi et al. [190] proved that a DL-CAD based on CNN and a deep polynomial network achieved better results in terms of accuracy (93.4 and 92.40%) and AUC (94.7%), respectively. In the same way, DL-CAD reached higher values than ML-CAD when used on private US images. For example, Shan et al. [35] and Singh et al. [41] showed ML based on an ANN for segmentation and classification that reached accuracies of 78.5 and 95.86% and an AUC of 82%, respectively. These works demonstrate that in most cases, the DL architectures outperformed traditional methodologies.

To conclude, the use of DL could be a promising new technique to obtain the main features for automatic breast tumor classification, especially in dense breasts. Furthermore, in medical image analysis, using DL has proven to be better for researchers compared to a conventional ML approach [41,42]. It appears as though DL provides a mechanism to extract features automatically through a self-learning network, thus boosting the classification accuracy. However, there is a continuing need for better architectures, more extensive datasets that overcome class imbalance problems, and better optimization methods.

Finally, the main limitation in this work is that several algorithms and results are not available in the open literature because of proprietary intellectual property issues.

**Author Contributions:** Conceptualization, Y.J.-G. and V.L.; methodology, Y.J.-G.; formal analysis, Y.J.-G., M.J.R.-Á., and V.L.; investigation, Y.J.-G.; resources, Y.J.-G.; writing—original draft preparation, Y.J.-G.; writing—review and editing, Y.J.-G., M.J.R.-Á., and V.L.; visualization, Y.J.-G.; supervision, M.J.R.-Á. and V.L.; project administration, M.J.R.-Á. and V.L.; funding acquisition, V.L. All authors have read and agreed to the published version of the manuscript.

**Funding:** This project has been co-financed by the Spanish Government Grant PID2019-107790RB-C22, "Software development for a continuous PET crystal systems applied to breast cancer".

**Conflicts of Interest:** The authors declare no conflict of interest.

## Abbreviations

| | |
|---|---|
| ANN | artificial neural network |
| CADx | computer-aided diagnosis |
| CADe | computer-aided detection |
| CNN | convolutional neural network |
| DM | digital mammography |
| DL | deep learning |
| DNN | deep neural network |
| DL-CAD | deep learning CAD system |
| CC | craniocaudal |
| MC | microcalcifications |
| ML | machine learning |
| MLO | mediolateral oblique |
| ROI | region of interest |
| US | ultrasound |
| MLP | Muli-layer perceptron |
| DBT | digital breast tomosynthesis |
| MIL | multiple instances learning |
| CRF | conditional random forest |
| RPN | region proposal network |
| GAN | generative adversarial network |
| IoU | intersection over union |
| SDAE | stacked denoising auto-encoder |
| CBIS | Curated Breast Imaging Subset |
| YOLO | You Only Look Once |
| ERFNet | Efficient Residual Factorized Network |
| CLAHE | contrast-limited adaptive histogram equalization |
| PCA | principal component analysis |
| LDA | linear discriminant analysis |
| GLCM | grey-level co-occurrence matrix |
| RF | random forest |
| DBT | decision boundary features |
| SVM | support vector machine |
| NN | neural network |
| SOM | self-organizing map |
| KNN | K-nearest neighbor |
| BDT | binary decision tree |
| DBN | deep belief networks |
| WPT | wavelet packet transform |

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
