# Peer review of "Deep-Learning-Based Computer-Aided Systems for Breast Cancer Imaging: A Critical Review"

_applsci, doi:10.3390/app10228298_

Round 1
Reviewer 1 Report
The manuscript consists on a review on CAD and Deep learning applications for breast cáncer diagnosis. It is bases on the literatura of the past decade. The paper is interesting. Authors should include and comment some of the most significant historic works from the beginning of CAD.
The manuscript is well written and organized. Authors conclude that the use of DL could be a promising new technique for authomatic breast tumor classification. AI is an important area of reseach in Radiology and a review of the methods are of interest for the readers.
This paper is in general well written. And It is potentially valuable. There are, however, a number of issues that should be addressed.
The introduction is clear. Some of the historic momments of CAD should be included. There should be a section with all the abbreviations. The key points of interest for the article have been highlighted.
In methods, data base that were utilized have been explained. Also, inclusion and exclusión criteria. PRISMA flow diagram is included.
In the description of the databases It should be explained, if possible, whether they are for detection o classification datasets.
Results. Table 5 and 6 and 7 should be simplified. A table with the best results in sensitivity and false positives is needed.
An evaluation of the methodologic quality of the studies should be done.
Discussion. I suppose that DM is mammography. US is complementará yo that. Both are necessary for the diagnosis.
A differentiation should be addressed between the methods that are already commercially available and those that are primary scientific research.
An evaluation of the methodologic quality of the studies should be done.
Authors should identify the "unknown information, data, etc" that still exist in the different CAD and DL systems. Is DL really superior to CAD? Limitations of the work should also be explained.
Figures. Authors should provide some figures with an example performing a CAD and DL schemes.
Author Response
Comment 1
The manuscript reviews CAD and Deep learning applications for breast cancer diagnosis. It is based on the literature of the past decade. The paper is interesting. Authors should include and comment some of the most significant historic works from the beginning of CAD.
Page 10
Line 264
R: The most significant ML-CAD traditional works are described in the table 2, as suggested.
Comment 4
There should be a section with all the abbreviations.
Page 1
Line 30
R: Some abbreviations section are included before the abstract section
Comment 6
In the description of the databases It should be explained, if possible, whether they are for detection o classification datasets.
Page 7
Line 30
- Done.
Comment 7
Results. Table 5 and 6 and 7 should be simplified. A table with the best results in sensitivity and false positives is needed.
Page 19
Line 632 table 5
Line 1022 table 6
Line 1178 table 7
Line 1189 table 8
- Simplified as suggested. Tables 6,7 and 8 contains sensitivity and specificity results.
Comment 8
An evaluation of the methodologic quality of the studies should be done.
Page
Line 1022 table 6
Line 1178 table 7
Line 1189 table 8
- The tables 6,7 and 8 show methodologic quality of the studies
Comment 9
Discussion. I suppose that DM is mammography. US is complementará yo that. Both are necessary for the diagnosis.
Page
R: Sorry, the comment is not clear enough.
Comment 10
A differentiation should be addressed between the methods that are already commercially available and those that are primary scientific research.
R: All of the results are from scientific research and not from commercially available software
Comment 11
An evaluation of the methodologic quality of the studies should be done.
Page
R: see response to comment number 8
Comment 12
Limitations of the work should also be explained.
Page 28
Line 1590
R: The main limitation in this work is that several algorithms and results are not available in the open literature, because of the proprietary intellectual property issues.
Comment 13
Figures. Authors should provide some figures with an example performing a CAD and DL schemes.
Page 9
Line 253
R: The figure 5 was added as an example of an CAD and DL-CAD scheme

Reviewer 2 Report
Presented paper is a compilation of experimental results from the literature. I found authors' contribution to the topic to be very low. Title promises "a critical review", while I find no criticism at all, just a bare presentation of methods, and some words of summary, which I wouldn't even call a discussion.
First of all the paper is hard to read due to excess of shortcuts (ML, DL, DM, US, etc.). Tables 1, 2, 5, 6, and 7 are barely possible to read. Their rows should rather be subchapters, there is no point putting so much text in the table. Why Table 3, and 4 are before Table 2? Formulas in tables 3, and 4 are corrupted.
Another aspect of the paper, which I don't like, is putting together things like in "Different breast image classification methods have been used to assist doctors in reading and interpreting medical images, such as Machine learning (ML), Deep learning (DL), and Computer-aided diagnosis/detection 71 (CAD) systems [8, 30-32].". Machine learning and/or deep learning systems may be a part of computer aided diagnosis systems. CADs are a use case of ML/DL systems. Similar statement is in "Also, there are some techniques for improving the CNNs performance such us Dropout, Batch normalization and Cross-validation.". While dropout and batch normalization may be considered as a method of improving performance, the cross validation is a method of assessing the performance (in means of accuracy).
There are also some unclear phrases: "DL techniques have been implemented to train neural networks in breast lesions detection, include an ensemble of CNN [69] and transfer learning [96,137,153].", "DL models produce a set of transformation functions and image features directly from the data [131], whose main advantage is to carry the burden of designing the specific features and the classification.", "That is why, researchers are using DL methods especially CNNs, because it methods have excellent results on segmentation task.", "Two strategies have been utilized in full image size for training CNN ond DM and US instead of ROIs. 1) High resolution [112] images and 2) patch-level [113].", "In the case of cancerous images, we need the lesion part and from it extracts its features.".
Finally, I find it unnecessary to describe in such details the quality measures. They are all well-defined in the literature, and I see no point of dedicating so much space for them.
Author Response
Comment 1
Presented paper is a compilation of experimental results from the literature. I found authors' contribution to the topic to be very low. Title promises "a critical review", while I find no criticism at all, just a bare presentation of methods, and some words of summary, which I wouldn't even call a discussion.
Page 27
Line 1290
R: We improve the discussion section including additional citations and further elaborated on the discussion of our results.
Comment 2
First of all the paper is hard to read due to excess of shortcuts (ML, DL, DM, US, etc.). Tables 1, 2, 5, 6, and 7 are barely possible to read. Their rows should rather be subchapters, there is no point putting so much text in the table. Why Table 3, and 4 are before Table 2? Formulas in tables 3, and 4 are corrupted.
Pages 1, 18-27,17
- Some abbreviations before abstract were included. The tables were summarized. The table 2, 3 and 4 were listed correctly. The equations were written again with Mathtype.
Comment 3
Another aspect of the paper, which I don't like, is putting together things like in "Different breast image classification methods have been used to assist doctors in reading and interpreting medical images, such as Machine learning (ML), Deep learning (DL), and Computer-aided diagnosis/detection (CAD) systems [8, 30-32]."
Page 3
Line 92
- The references were classified separately according to the DL, ML and CAD methods and corrected as suggested.
Comment 4
Similar statement is in "Also, there are some techniques for improving the CNNs performance such us Dropout, Batch normalization and Cross-validation.". While dropout and batch normalization may be considered as a method of improving performance, the cross validation is a method of assessing the performance (in means of accuracy).
Page 18
Line 398
- There was a mistake to consider a Cross-validation method for improving the CNN performance, and it was corrected as suggested.
Comment 5
There are also some unclear phrases: "DL techniques have been implemented to train neural networks in breast lesions detection, include an ensemble of CNN [69] and transfer learning [96,137,153].", "DL models produce a set of transformation functions and image features directly from the data [131], whose main advantage is to carry the burden of designing the specific features and the classification.", "That is why, researchers are using DL methods especially CNNs, because it methods have excellent results on segmentation task.", "Two strategies have been utilized in full image size for training CNN ond DM and US instead of ROIs. 1) High resolution [112] images and 2) patch-level [113].", "In the case of cancerous images, we need the lesion part and from it extracts its features.".
Page 15
Line 474
- The phrases were corrected as suggested
DL techniques have been implemented to train neural networks in breast lesions detection, including ensemble [69] and transfer learning [96,137,153] methods.
Line 438
- DL models produce a set of image features from the data [131], whose main advantage is that they extract features and perform classification directly
Line 382
- Two strategies have been utilized in full image size for training CNN on DM and US instead of ROIs. 1) High resolution [112] images and 2) patch-level [113]
Line 340
- line was deleted
Comment 6
I find it unnecessary to describe in such details the quality measures. They are all well-defined in the literature, and I see no point of dedicating so much space for them.
Page 18
R: The quality measures were eliminated, as suggested, and put into table

Round 2
Reviewer 1 Report
Ser Comments to editores
Reviewer 2 Report
Changes made to the paper fulfill (most of) my comments.